# Impact of Goal Directed Therapy in Head and Neck Oncological Surgery with Microsurgical Reconstruction: Free Flap Viability and Complications

**DOI:** 10.3390/cancers13071545

**Published:** 2021-03-27

**Authors:** Blanca Tapia, Elena Garrido, Jose Luis Cebrian, Jose Luis Del Castillo, Javier Gonzalez, Itsaso Losantos, Fernando Gilsanz

**Affiliations:** 1Anesthesia and Intensive Care Department, University Hospital La Paz, Universidad Autónoma de Madrid, 28046 Madrid, Spain; fernando.gilsanz@salud.madrid.org; 2Anesthesia and Intensive Care Department, Wexner Medical Center, 410 W 10th Ave, Columbus, OH 43210, USA; elena.garrido@osumc.edu; 3Oral and Maxillofacial Surgery Department, University Hospital La Paz, Universidad Autónoma de Madrid, 28046 Madrid, Spain; josel.cebrian@salud.madrid.org (J.L.C.); josedel.castillo@salud.madrid.org (J.L.D.C.); jgmartinmoro@salud.madrid.org (J.G.); 4Statistics Department, Hospital La Paz, Paseo de la Castellana, 261, 28046 Madrid, Spain; itsaso.losantos@salud.madrid.org

**Keywords:** goal directed therapy, free flap surgery, fluid therapy, cardiac output monitors

## Abstract

**Simple Summary:**

Based on the proven benefits of goal directed therapy (GDT) in the perioperative management of different surgical procedures and in high-risk patients, we hypothesised that this approach would also be beneficial in microvascular free flap reconstruction in head and neck cancer. In this study, we investigated whether GDT would directly benefit flap viability in addition to improving morbidity and mortality. As this reconstructive technique is gradually being introduced in more specialist fields, particularly radical oncological surgery, the benefits of GDT in this context could be extended to numerous procedures.

**Abstract:**

(1) Background: Surgical outcomes in free flap reconstruction of head and neck defects in cancer patients have improved steadily in recent years; however, correct anaesthesia management is also important. The aim of this study has been to show whether goal directed therapy can improve flap viability and morbidity and mortality in surgical patients. (2) Methods: we performed an observational case control study to analyse the impact of introducing a semi invasive device (Flo Trac^®^) during anaesthesia management to optimize fluid management. Patients were divided into two groups: one received goal directed therapy (GDT group) and the other conventional fluid management (CFM group). Our objective was to compare surgical outcomes, complications, fluid management, and length of stay between groups. (3) Results: We recruited 140 patients. There were no differences between groups in terms of demographic data. Statistically significant differences were observed in colloid infusion (GDT 53.1% vs. CFM 74.1%, *p* = 0.023) and also in intraoperative and postoperative infusion of crystalloids (CFM 5.72 (4.2, 6.98) vs. GDT 3.04 (2.29, 4.11), *p* < 0.001), which reached statistical significance. Vasopressor infusion in the operating room (CFM 25.5% vs. GDT 74.5%, *p* < 0.001) and during the first postoperative 24h (CFM 40.6% vs. GDT 75%, *p* > 0.001) also differed. Differences were also found in length of stay in the intensive care unit (hours: CFM 58.5 (40, 110) vs. GDT 40.5 (36, 64.5), *p* = 0.005) and in the hospital (days: CFM 15.5 (12, 26) vs. GDT 12 (10, 19), *p* = 0.009). We found differences in free flap necrosis rate (CMF 37.1% vs. GDT 13.6%, *p* = 0.003). One-year survival did not differ between groups (CFM 95.6% vs. GDT 86.8%, *p* = 0.08). (4) Conclusions: Goal directed therapy in oncological head and neck surgery improves outcomes in free flap reconstruction and also reduces length of stay in the hospital and intensive care unit, with their corresponding costs. It also appears to reduce morbidity, although these differences were not significant. Our results have shown that optimizing intraoperative fluid therapy improves postoperative morbidity and mortality.

## 1. Introduction

Microvascular free flaps are now the preferred method of reconstruction for most major head and neck cancer defects, giving better functional and cosmetic outcomes and generally higher success rates compared to local and regional flaps [1].

With the introduction of more advanced microvascular instruments and the refinement of microvascular techniques, free flap surgery has become a reliable and efficient method for reconstructing complex defects [2,3,4]. Despite reported free flap survival rates of 95% to 99%, the possibility of flap failure is still a major concern [4]. The free flap is transferred with its accompanying artery and vein and reattached to vessels at the donor site using microvascular techniques. Good intraoperative anaesthetic management is considered critical in achieving good flap outcome [5,6].

There is a growing body of evidence to recommend the use of cardiac output monitors in high-risk patients or patients undergoing major surgery [7,8]. Haemodynamic monitoring together with blood pressure-based perfusion management strategies can reduce morbidity, mortality, and length of stay in both the intensive care unit (ICU) and the ward, resulting in savings to the healthcare system [9]. It is also useful to bear in mind that haemodynamic monitoring is particularly important in patients with generalised hypoperfusion undergoing microvascular free flap reconstruction [10]. Goal directed therapy (GDT) involves administering fluids and vasoactive drugs according to haemodynamic objectives [10]. It was originally used in surgical patients in whom normal or supranormal cardiac output and oxygenation values were required to meet increased perioperative oxygen demands and prevent organ failure. Individualizing fluid management means tailoring fluid administration to individual needs by ensuring that the patient’s heart is operating close to the inflection point on the Frank Starling curve, far from the dangerous hypovolemic and fluid overload zones [11].

This study was designed to evaluate the benefit of using a goal directed algorithm based on fluid management recommendations for this group of cancer patients.

The primary outcome measure was improved flap survival. Secondary outcome measures were reduced morbidity and mortality associated with the procedure and shorter hospital and intensive care unit stay.

Monitoring was performed with Flo Trac ^®^, a system that uses an algorithm based on the principle that aortic pulse pressure is proportional to stroke volume (SV) and inversely related to aortic compliance. In addition, compliance inversely affects pulse pressure (PP), as the algorithm compensates for the effects of compliance on PP based on age, gender, and body surface area.

Our findings show that goal directed therapy in free flap reconstruction in head and neck cancer improves free flap survival and reduces the morbidity and mortality associated with the procedure. This approach can, therefore, be recommended in this context.

Similarly, the administration of hydroxyethyl starch and vasopressors during the perioperative period does not increase the risk of 30-day complications or the risk of flap necrosis.

## 2. Results

### 2.1. Demographic Data

There were no differences between groups in terms of demographic data, disease history, or background therapy, as shown in Table 1, Table 2 and Table 3.

Groups were evenly matched in terms of America Society of Anesthesiology (ASA) classification:

ASA I (14.4% vs. 20.3%) ASA II (62.3% vs. 54.2%) ASA III (21.7 vs. 22%), ASA IV (1.4% vs. 3.4%) in groups conventional fluid management (CFM) vs. GDT, respectively.

Anticoagulant or antiplatelet therapy did not differ between groups, and transfusion rates were also similar (*p* = 0.24).

### 2.2. Surgical Data

The following results were obtained in different types of flap, with no significant differences between groups (Table 4.)

The percentage of bone grafts with respect to soft tissue grafts is higher in the CFM group, *p* = 0.039.

More patients in the CFM group underwent tracheotomy (76.8%) compared to the GDT group (57.6%) (*p* = 0.024).

The number of patients undergoing neck dissection (unilateral or bilateral) did not differ statistically between groups, *p* = 0.341.

Surgical time was also similar in both groups. CFM: median 600 (320,900) vs. GDT 582 (400,1200). *p* = 0.638 (minutes).

### 2.3. Fluid Management and Vasopressor Data

Despite differences in the percentage of patient receiving colloids in each group (CFM 53.1% vs. GDT 74.1%, *p* = 0.023), there were no statistically significant differences in the volume of colloids administered (Figure 1). However, intraoperative and postoperative infusion of crystalloids (CFM 5.72 (4.2, 6.98) vs. GDT 3.04 (2.29, 4.11), *p* < 0.001) differed significantly between groups (Figure 2 and Figure 3).

Vasopressor infusion in the operating room (CFM 25.5% vs. GDT 74.5%, *p* < 0.001) and during the first 24 h (CFM 40.6% vs. GDT 75%, *p* > 0.001) also differed. Our results show that the use of vasopressors and perioperative administration of colloids does not affect flap prognosis or the presence of 30-day complications (particularly renal failure in the first 24 h and at 30 days) (Table 5). The table shows the effect of colloids and vasopressors on different events, in other words, the percentage of patients presenting bleeding or flap thrombosis that received colloids and vasopressors and the percentage that did not (Table 6).

### 2.4. Length of Stay

Differences were also found in length of stay in the intensive care unit (hours: CFM 58.5 (40, 110) vs. GDT 40.5 (36, 64.5), *p* = 0.005) (Figure 4) and in the hospital (days: CFM 15.5 (12, 26) vs. GDT 12 (10, 19), *p* = 0.009) (Figure 5).

### 2.5. Postoperative Results

Nearly twice as many CFM patients presented one or several complications compared to GDT patients (CFM 28.6% vs. GDT 13.7%, *p* = 0.07), although these differences were not statistically significant. Bleeding complications (CFM 21% vs. GDT 18.9%, *p* = 0.82), and kidney disease during hospital stay (CFM 11.3% vs. GDT 22.2%, *p* = 0.1) or at 30 days (CFM 4.7% vs. GDT 12.5%, *p* = 0.18) did not differ between groups.

The free flap necrosis rate differed between groups (CFM 37.1% vs. GDT 13.6%, *p* = 0.003) (Figure 6). Partial necrosis is defined as flaps that became viable after the surgical review. An analysis of the viability of these flaps (after surgical review) showed no significant differences (CFM 88.2% vs. GDT 94.5%).

One-year survival did not differ between groups (CFM 95.6% vs. GDT 86.8%, *p* = 0.08).

## 3. Discussion

The results of our study are similar to those reported in surgery involving microvascular anastomoses and the findings of studies in high surgical risk patients. However, unlike previous studies, our analysis opens a new field of study in the use of goal directed therapy in free flap surgery that has not hitherto been explored.

For the anaesthesiologist, the objective is to achieve good tissue oxygenation by maintaining adequate cardiac output to optimize flap perfusion [12]. Insufficient fluid replacement in the event of hypovolaemia reduces cardiac output (CO) and decreases oxygen delivery (DO2) to the free flap, resulting in graft failure. However, a positive fluid balance can also be harmful. Flaps have a high risk of developing oedema due to a lack of lymphatic drainage, denervation, and poor reabsorption of excess interstitial fluid [13]. Earlier studies have recommended administering less than 6 mL/kg/h for maintenance, but the latest research has focused on the use of advanced haemodynamic monitoring and the benefit of GDT to improve outcomes in critical patients and those undergoing high-risk surgery [14]. The question of when to use colloids or crystalloids, however, remains unclear. Firstly, infusion solutions are obviously drugs and as such have indications, contraindications, and side effects [15]. Therefore, current “safety” discussions are misdirected, because they ignore the fact that any potent drug can only be administered after carefully weighing up the pros and cons on an individual basis. Secondly, colloids and crystalloids are completely different classes of drugs with different pharmacokinetics and target compartments [16]. In our study, we have been able to show that the use of colloids is not harmful in the short and long term, flap survival is not affected by the use or absence of colloids, and the 30-day complication rate was not higher in the group receiving hydroxyethyl starch. Chan et al. reported in 1983 that surgical trauma produces oedema in the injured tissue. These authors showed that the creation of small bowel anastomosis in rabbits increased tissue weight by 5%–10% due to fluid accumulation. Supplementary intravenous crystalloid infusion of 5 mL/kg/h doubled the size of the oedema and destabilized the anastomosis [17,18]. Noblett et al. randomized 108 patients undergoing colorectal resection to intraoperative GDT or standard fluid management (3638 mL vs. 3834 mL), and showed that GDT significantly reduced interleukin 6 levels. Therefore, using GDT to ensure splanchnic circulation reduced the systemic inflammatory response due to surgical trauma [19]. In GDT, the administration of fluids and vasopressors is guided by targets such as cardiac index, stroke volume, stroke volume variation, or oxygen delivery representing end organ blood flow. The conclusions of a recent metanalysis show that current perioperative goal directed therapy reduced mortality and morbidity, although the quality of evidence is low and there is considerable clinical heterogeneity among goal directed therapy devices and protocols currently in use [20,21]. UK guidelines consider GDT to be a quality factor in anaesthesia management in head and neck surgery. Ettinger et al. showed that intraoperative fluid volume is significantly associated with the risk of postoperative complications in head and neck reconstruction surgery. In a retrospective review, the authors found fluid volume greater than 5500 mL to be significantly associated with complications [22,23,24].

GDT using a non-calibrated pulse wave analysis device (FloTraq ^®^, Edwards Lifesciences, Irvine, CA, USA) [25] showed clinical benefits in several general surgical cohorts [26]. Free flap blood flow decreases to around half pretransfer values during the first 6 to 12 postoperative hours [27]; nevertheless, use of vasoactive agents during free flap surgery is still controversial. Flap perfusion can be improved by increasing systemic blood pressure with vasoconstrictive agents; however, this has never been demonstrated in prospective clinical trials [28]. Systemic phenylephrine increases systemic vascular resistance and arterial pressure by 30% and appears to have no adverse effects on blood flow in free musculocutaneous flaps. This drug is, therefore, the first choice in our algorithm if vasopressors are required.

A multivariant analysis on the use of vasopressors during reconstructive surgery in 139 patients did not show statically significant differences in flap complications compared with nonrecipients [29]. Previous reports provide no clinical evidence against the selective use of vasoactive agents during free flap surgery, a finding that has been confirmed in our study.

In 2017, the Enhanced Recovery After Surgery (ERAS) Society endorsed the development of the first ERAS protocol for patients undergoing head and neck cancer surgery with free flap reconstruction. The strategy included a patient diary, nutritional optimization, tracheostomy avoidance, intraoperative goal directed fluid therapy, and a specific postoperative head and neck pain management protocol [30].

On the basis of these hypotheses, and supported by the ERAS recommendations, we have confirmed that goal directed therapy is extremely beneficial in both improving graft viability and reducing overall complications.

## 4. Materials and Methods

### 4.1. Study Design

After obtaining approval from the Ethics Committee of the Hospital de la Paz on 12 March 2015, under number PI-1967, patient recruitment was started for this single-centre, observational, non-crossover case control study. Cases from 2011 to 2014, before the introduction of the haemodynamic monitoring system (FloTraq^®^), were included retrospectively; the remaining patients were included prospectively up to 2018.

A total of 140 patients were recruited; graft prognosis data were missing in 11 cases, and other data, namely the anaesthesia report showing intraoperative fluid administration, the use, type, and dose of vasoactive agents required, and intraoperative diuresis, were missing in a further eight cases. All these cases were ultimately eliminated from the final analysis, leaving a total of 121 study patients. Both the surgical and anaesthesia team remained unchanged over the study period.

For the purpose of analysis, we reviewed the medical records of the CFM group, while subjects for the GDT group were recruited at the time of surgery. The same data were analysed from both groups; in the CFM group, they were collected retrospectively while in the GDT group they were collected prospectively.

In the CFM group, intraoperative fluid therapy was performed at the discretion of the two-attending anaesthesiologist, while in GDT patients it was performed according to an algorithm based on recommendations in this type of procedure. The values used in the algorithm were extracted from the hemodynamic monitoring system.

The goal directed therapy algorithm is described in Figure 7.

Inclusion criteria: All cancer patients scheduled for reconstructive surgery of the head and neck with a microvascular free flap between 2010 and the first quarter of 2018 were included in the study.

Exclusion criteria: Need for urgent surgery.

All patients were monitored according to the rules of the Spanish Society of Anaesthesia (SEDAR): electrocardiogram, pulse oximetry, non-invasive blood pressure (BP) during induction, after which a radial or femoral arterial line was placed for BP monitoring. In GDT patients, the same line was used to measure haemodynamic parameters. Depth of anaesthesia was monitored using the BIS (Bispectral Index Sensor).

Active warming systems (convective blanket and fluid warmer) were used in all patients, and temperature was measured in the GDT group.

Balanced general anaesthesia was performed in all patients. Anaesthesia was induced with 1.5–2 mg/kg propofol, 5–7 mcg/kg fentanyl, and 0.6–0.8 mg/kg rocuronium. Fibreoptic nasotracheal intubation was performed in patients (the majority) in whom the airway evaluation predicted the risk of a “cannot intubate, cannot ventilate” scenario. Anaesthesia was maintained with inhalation agents (sevoflurane) in both groups. After surgery, patients were transferred to the post-anaesthesia care unit (PACU) for a few hours before extubation until immediate complications (bleeding, suffering of the flap) had been ruled out.

### 4.2. Parameters Collected

Demographic dataAgeWeightSexPreoperative factorsSmokingHigh blood pressureChronic obstructive pulmonary diseaseAlcoholismDyslipidaemiaPrevious chemotherapyPrevious radiation therapyAcute myocardial infarctionHistory of arrhythmiaPresence of hypoalbuminemiaDiabetes mellitusASA classificationHaemoglobin level, presence of preoperative anaemiaBackground medication: antidepressants, benzodiazepines, antihypertensives (CC blockers, ACE inhibitors/ARBs, beta blockers)Anticoagulants/antiplatelet agentsIntraoperative factorsUse or non-use of FloTraq^®^Type of flapBone/non-bone flapSurgical timeNeed for transfusion (volume)TracheotomyNeck dissectionFluid therapyCrystalloidsColloidsVasopressorsPostoperative factorsPACU length of stay (hours)Hospital length of stay (days)Time on ventilator (hours)Twenty-four hour fluid therapyCrystalloidsColloidsNeed for transfusionVasopressorsPostoperative complicationsBleedingThrombosisOther complicationsPneumoniaArrhythmiaIschaemic heart diseasePulmonary thromboembolismDeep vein thrombosisCongestive heart failureConfusionCerebral vascular accident (CVA)Creatinine above 1.2 mg/mLCreatinine greater than 24 h per monthDiuresis first 24 hAntiplatelet/anticoagulant medication at the time of pedicle sectionHeparinLysine acelsalicylateAntithrombotic prophylaxis 12 h before the procedure40 IU low molecular weight heparin (LMWH)/noGraft outcomeNecrosisPartial (flaps rescued in a subsequent surgical review)TotalViabilitySurvivalYes/No

### 4.3. Sample Size Calculation

As the sample size to calculate flap necrosis would be too high, we decided to carry out a pilot study over a period of seven years.

### 4.4. Statistical Analysis

Qualitative data are presented as absolute frequencies and percentages and quantitative data as mean ± standard deviation (SD), minimum and maximum if they were normally distributed, and median and interquartile range if they were not.

The association between qualitative variables was analysed using the chi-square test or Fisher’s exact test.

The nonparametric Mann–Whitney test was used to compare qualitative and quantitative data.

Tests were two-tailed and a *p* value less than 0.05 was considered statistically significant. Statistical analysis was performed on SAS 9.3 (SAS Institute, Cary, NC, USA).

## 5. Conclusions

Goal directed therapy in cancer patients undergoing head and neck reconstruction surgery should be included in all perioperative management protocols, since it improves surgical outcomes and quality of recovery at discharge, and reduces both postoperative complications and length of hospital stay.

Goal directed therapy in head and neck cancer reconstruction surgery improves free flap outcomes and also reduces hospital and intensive care unit length of stay, with their corresponding costs. It also appears to reduce morbidity, although these differences were not significant. Our results have shown that optimizing intraoperative fluid therapy improves postoperative morbidity and mortality.

We are considering continuing recruitment in order to minimise the possibility that differences between bone/non-bone grafts could be a source of bias in our study.

## Figures and Tables

**Figure 1 cancers-13-01545-f001:**
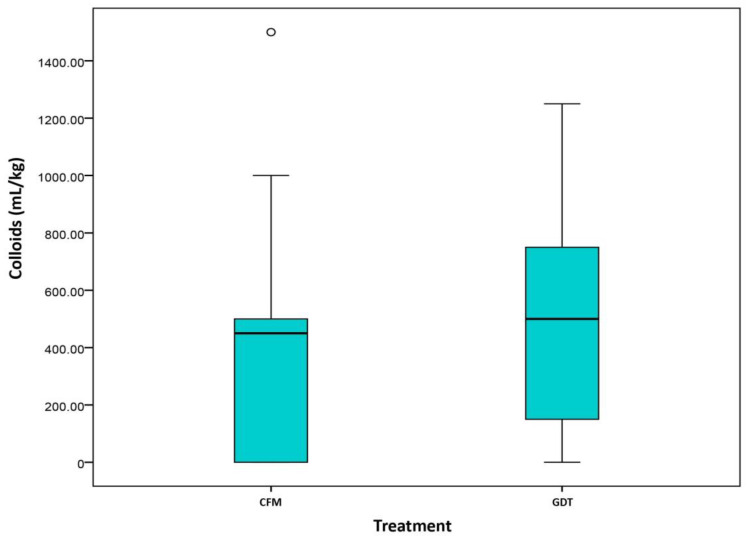
Colloids (Day 1). Classical fluid management (CMF)/Goal directed therapy (GDT).

**Figure 2 cancers-13-01545-f002:**
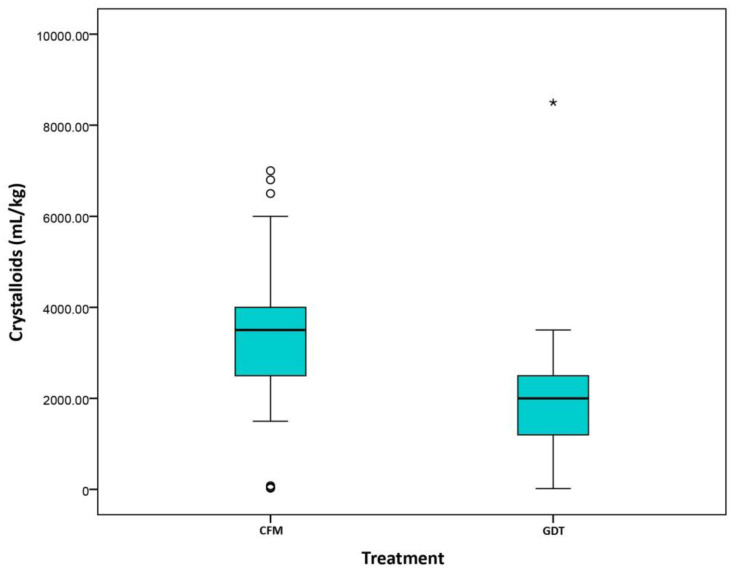
Crystalloids (mL/kg) (Day 1). Classical fluid management (CMF)/Goal directed therapy (GDT).

**Figure 3 cancers-13-01545-f003:**
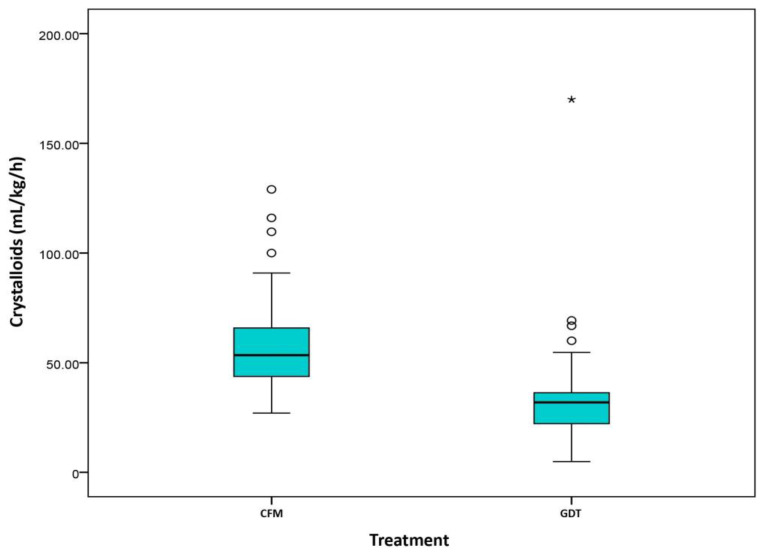
Crystalloids (mL/kg/h) (Day 1). Classical fluid management (CMF)/Goal directed therapy (GDT).

**Figure 4 cancers-13-01545-f004:**
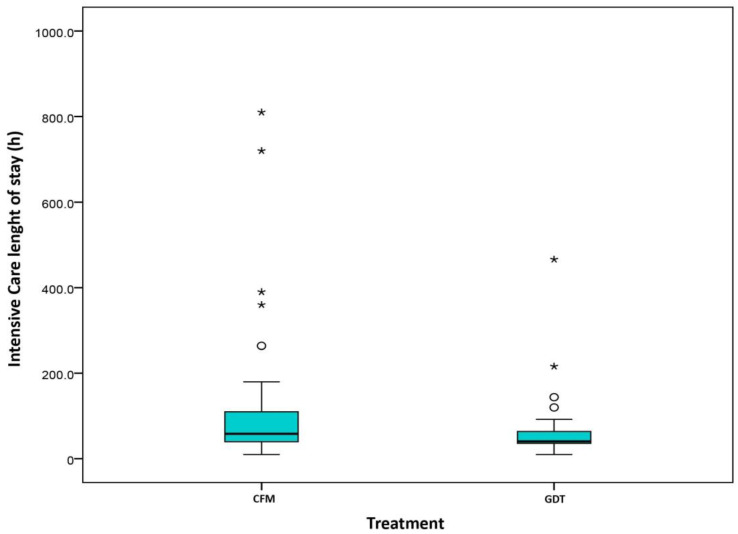
Intensive care length of stay (hours). Classical fluid management (CMF)/Goal directed therapy (GDT).

**Figure 5 cancers-13-01545-f005:**
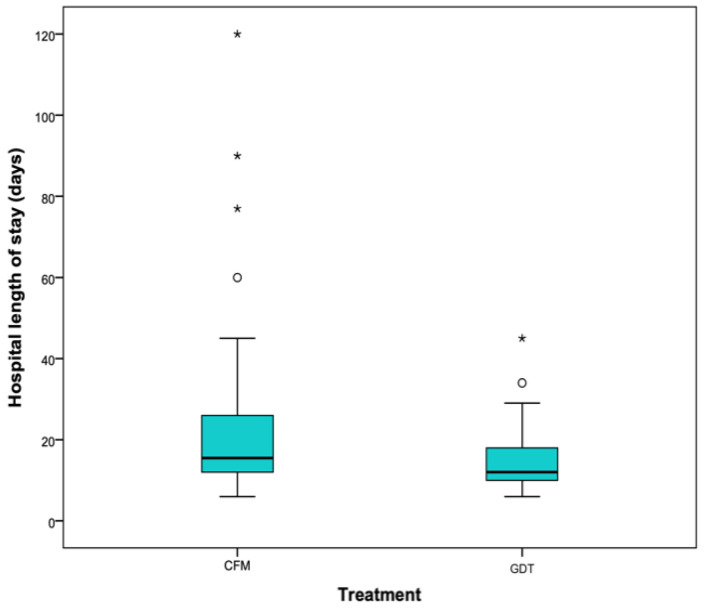
Hospital length of stay (days). Classical fluid management (CMF)/Goal directed therapy (GDT).

**Figure 6 cancers-13-01545-f006:**
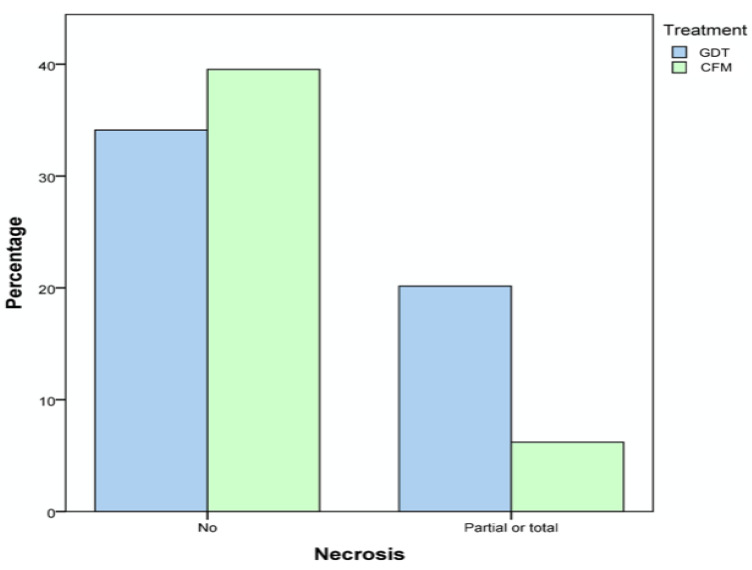
Free flap necrosis rate (%).

**Figure 7 cancers-13-01545-f007:**
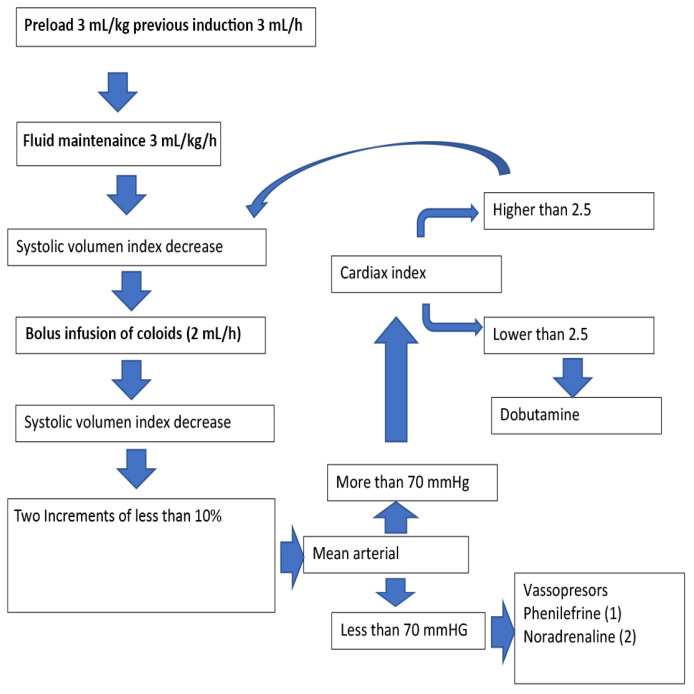
GDT algorithm.

**Table 1 cancers-13-01545-t001:** Demographic Data.

	Conventional Fluid Management (CFM)	Goal Directed Therapy (GDT)
Gender (male) %	49.1	57.9
Age, years (median, interquartile range)	58 (18/87)	58.50 (18/81)
Weight, kilograms (median, interquartile range)	65 (45/95)	65 (46/127)

**Table 2 cancers-13-01545-t002:** Disease history.

	CFM	GDT
Smoker (%)	51.9	48.1
Hypertension (%)	48.9	51.1
Chronic pulmonary disease (%)	51.5	48.5
Alcoholism (%)	55.3	44.7
Dyslipidaemia (%)	52	48
Ischaemic heart disease (%)	1.4	8.5
Arrhythmia (%)	1.4	5.2
Hypoalbuminemia (%)	4.3	3.4
Diabetes mellitus (%)	1.4	6.8
Anaemia (%)	17.9	22.8

**Table 3 cancers-13-01545-t003:** Background therapy.

	CFM	GDT
Antidepressants (%)	11.6	8.5
Benzodiazepines (%)	23.2	18.6
Calcium channels (CC) blockers (%)	4.3	10.2
Angiotensin-converting enzyme (ACE) inhibitors)/Angiotensin II receptor blockers (ARBs) (%)	19.1	33.9
Beta-blockers (%)	8.7	6.9

**Table 4 cancers-13-01545-t004:** Free flap types (percentage).

	CFM	GDT
Anterolateral thigh	25%	21.7%
Radial/cubital/forearm	29.2%	41.7%
Fibula	36.1%	21.7%
Iliac crest	4.2%	3.3%
Scapula bone	1.4%	0%
Others	4.1%	11.6%

**Table 5 cancers-13-01545-t005:** Colloids and complications.

	Yes	No	*p* Value
Free flap bleeding (%)	12.5	52.4	*
Free flap thrombosis (%)	75.8	90	*
Medical/surgical complications (%)	25.6	20.3	0.633
Renal failure (first 48h/30 days)	14.7/13	14.1/3.2	*/*

* We were unable to obtain a *p*-value due to the low number of cases in both groups.

**Table 6 cancers-13-01545-t006:** Vasopressors and complications.

	Yes	No	*p* Value
Free flap bleeding (%)	42.9	46.7	1
Free flap	78,6	100	*
thrombosis (%)			
Medical/surgical complications (%)	25.4	20.5	0.641
Renal failure (%)	15.9/6.5	14/9.8	1/*
(24 h/30 days)			

* We were unable to obtain a *p*-value due to the low number of cases in both groups.

## Data Availability

The data presented in this study are available on request from the corresponding author.

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
