# Peer review of "Impact of Goal Directed Therapy in Head and Neck Oncological Surgery with Microsurgical Reconstruction: Free Flap Viability and Complications"

_cancers, 2021, doi:10.3390/cancers13071545_

Round 1
Reviewer 1 Report
The study design is through and the conclusions drawn from this study are helpful.
Few details to address.
The reference style is not consistent, spacing before or after reference.
Paragraph organization can be made better. The results description can be improved which reads like a list of reported numbers.
Figures: The axis labels are not clear and the font is too small.
Author Response
Point 1
The study design is through and the conclusions drawn from this study are helpful.
Few details to address.
The reference style is not consistent, spacing before or after reference.
References style, has been changed as per your comments
.Point 2
Paragraph organization can be made better.
Paragraph organization has been revised in order to make it easier for the reader
Point 3
The results description can be improved which reads like a list of reported numbers.
In order to clarify this point; The structure of the document has been revised being the outcome an easy read
.Point 4
Figures: The axis labels are not clear and the font is too small.
Fig 1, 2 3 4 and 5 have been modified in order to simplify the understanding of the document
Point 5
Fig 7 and line 118 has been revised in order to correct some mistake detected by me
Reviewer 2 Report
Great paper. Well-written. This will add to the body of existing literature and provides important data for microvascular surgeons.
Author Response
Great paper. Well-written. This will add to the body of existing literature and provides important data for microvascular surgeons.
Thank you very much for your nice words.
Fig 7 and line 118 has been revised in order to correct some mistake detected by me
Reviewer 3 Report
This is a generally well written manuscript on a relevant subject.
My main concern is that the number of bone flap with less reliable skin paddles (mainly fibula) is significantly higher in the CFM group - that represents a major bias as flap survival was used as the the main outcome measure. I do like the general design of the study but this is a problem. It would be worthwhile to prologue the study and add some fibula flaps to the CFM group and the other way round!
minor comments:
Abstract: The number of patients effectively used in the study should be mentioned instead of the theoretical 140. In the Methods you mention 129 patients and then mention that 8 were excluded. Does that mean that only 121 were effectively included? Somehow the numbers don't add up.
The numeration of tables is a mess, the "4" is missing in Table 4 and the following tables have the wrong numbers (also in the text). Also % symbols are missing in Table 4.
Figure 2 should be rotated 90 degrees to the left.
Author Response
This is a generally well written manuscript on a relevant subject.
Point 1
My main concern is that the number of bone flap with less reliable skin paddles (mainly fibula) is significantly higher in the CFM group - that represents a major bias as flap survival was used as the the main outcome measure. I do like the general design of the study but this is a problem. It would be worthwhile to prologue the study and add some fibula flaps to the CFM group and the other way round!
The conclusion of the document and the result of the study are extracted from the patients analysed
As per you suggestion I have included in the conclusion a paragraph explaining the need to increase the number of patients:
“We are considering continuing recruitment in order to minimise the possibility that differences between bone/non-bone grafts could be a source of bias in our study. “line 350
minor comments:
Point 2:
Abstract: The number of patients effectively used in the study should be mentioned instead of the theoretical 140. In the Methods you mention 129 patients and then mention that 8 were excluded. Does that mean that only 121 were effectively included? Somehow the numbers don't add up.
I have changed the text for the next paragraph. I hope it will be easy to understand
“”A total of 140 patients were recruited; graft prognosis data were missing in 11 cases, and other data, namely the anaesthesia report showing intraoperative fluid administration, the use, type and dose of vasoactive agents required, and intraoperative diuresis, were missing in a further 8 cases. All these cases were ultimately eliminated from the final analysis, leaving a total of 121 study patients”. (line 261)
Point 3
The numeration of tables is a mess, the "4" is missing in Table 4 and the following tables have the wrong numbers (also in the text). Sorry for these misunderstandings, they were type mistakes.
Point 4:
They have been corrected Also % symbols are missing in Table 4. Already included.
Point 5
Figure 2 should be rotated 90 degrees to the left.
Yes, I have rotated, and I´ve also modified the rest of the figures for an easy reading.
Point 6
Fig 7 and line 118 has been revised in order to correct some mistake detected by me
Round 2
Reviewer 3 Report
Thank you for your corrections. I am still a little bit concerned about the potential bias due to the different types of flaps but as you have now mentioned it as a potential confounder the readers can make up their own mind and I am willing to accept it as it is.